# Microbe Interactions within the Skin Microbiome

**DOI:** 10.3390/antibiotics13010049

**Published:** 2024-01-04

**Authors:** Thaís Glatthardt, Rayssa Durães Lima, Raquel Monteiro de Mattos, Rosana Barreto Rocha Ferreira

**Affiliations:** 1Instituto de Microbiologia Paulo de Góes, Universidade Federal do Rio de Janeiro, Rio de Janeiro 21941-902, Brazil; thais.glatthardtdasi@ucalgary.ca (T.G.); rayssalima@ku.edu (R.D.L.); raquelmattos00@gmail.com (R.M.d.M.); 2Department of Physiology and Pharmacology, Health Research Innovation Centre, University of Calgary, Calgary, AB T2N 4N1, Canada; 3Department of Pediatrics, Alberta Children Hospital Research Institute, University of Calgary, Calgary, AB T2N 4N1, Canada; 4Department of Molecular Biosciences, The University of Kansas, Lawrence, KS 66045, USA

**Keywords:** microbial signaling, quorum sensing, skin microbiota, *Staphylococcus*, *Cutibacterium acnes*

## Abstract

The skin is the largest human organ and is responsible for many important functions, such as temperature regulation, water transport, and protection from external insults. It is colonized by several microorganisms that interact with each other and with the host, shaping the microbial structure and community dynamics. Through these interactions, the skin microbiota can inhibit pathogens through several mechanisms such as the production of bacteriocins, proteases, phenol soluble modulins (PSMs), and fermentation. Furthermore, these commensals can produce molecules with antivirulence activity, reducing the potential of these pathogens to adhere to and invade human tissues. Microorganisms of the skin microbiota are also able to sense molecules from the environment and shape their behavior in response to these signals through the modulation of gene expression. Additionally, microbiota-derived compounds can affect pathogen gene expression, including the expression of virulence determinants. Although most studies related to microbial interactions in the skin have been directed towards elucidating competition mechanisms, microorganisms can also use the products of other species to their benefit. In this review, we will discuss several mechanisms through which microorganisms interact in the skin and the biotechnological applications of products originating from the skin microbiota that have already been reported in the literature.

## 1. Introduction

The skin is the largest organ of our bodies and is an important interface where the internal and external environments interact. As such, the skin protects us against many external factors, such as chemicals, pathogens, and changes in temperature [1,2]. The skin is formed by three layers, termed the dermis, epidermis, and hypodermis, which are composed of lipid molecules, cornified cells, keratinocytes cells, and several invaginations, including sweat glands, hair follicles, and sebaceous glands [2]. The variation in density and number of glands and hair follicles along the skin creates completely distinct niches for microbial growth due to the complexity of physical and chemical conditions of each microenvironment [3]. These differences affect not only the abundance of microbes at each skin compartment but also the composition of the microbial communities found [4]. Furthermore, several factors (both intrinsic and extrinsic) can alter the microbiota composition, including aging, gender, ethnicity, hygiene habits, use of cosmetics, use of antibiotics, and geographic location [5].

Several studies have analyzed the composition of the skin microbiome according to specific sites on the skin, which have been divided into dry, sebaceous, and humid areas [6]. For example, in sebaceous sites, *Propionibacterium* (mostly belonging to the recently named *Cutibacterium* genus) [7] is the most prevalent genus, whereas *Staphylococcus* and *Corynebacterium* are the most common genera in moist areas [4]. Broadly, *Staphylococcus*, *Corynebacterium*, and *Cutibacterium acnes* are the most frequent bacteria found on the epithelial surface [6,8].

Similar to what happens at different body sites, a diverse skin microbiome is essential for healthy and well-functioning skin, and dysbiosis has been observed at the onset of skin diseases, such as atopic dermatitis and psoriasis [9]. Although the mechanisms behind the dysbiosis in these diseases are not well known, and there are ongoing discussions about whether dysbiosis is a cause or a consequence of the diseases, the skin microbiota exerts an important role in maintaining homeostasis. Similarly to what is found in the gut, the skin microbiome can protect the host against pathogens, a function known as colonization resistance [5]. The processes by which colonization resistance occurs are the focus of recent studies and involve not only competition for adhesion sites and nutrients but also signaling molecules. Understanding the mechanisms behind skin colonization resistance, and the molecules involved, will hopefully lead to alternatives for fighting bacterial infections, especially with the increasing antimicrobial resistance threat [10]. In this review, we cover recent data on the interactions between different commensals residing in the skin microbiota, as well as between commensals and human skin pathogens, with a focus on the molecules produced during such interactions and their potential role in skin colonization resistance.

## 2. Bioactive Molecules Produced during Microbial Interactions in the Human Skin

Overall, complex microbial communities are characterized by different interactions that shape the microbial structure and community dynamics. Regarding the human microbiome, different studies have focused on understanding these microbial interactions, especially in the gut microbiome, and have shown their role as an important mechanism that modulates microbiome balance and host health maintenance [11,12].

The skin is a nutrient-poor habitat directly exposed to the external environment and, consequently, exogenous microorganisms. This combination implies that there is a considerable amount of competition between resident microorganisms and between them and transient microorganisms [4,13]. Therefore, the ability to produce molecules that affect the establishment of other microorganisms in this type of environment confers a fitness advantage (Table 1).

One of the most studied competition mechanisms found in bacteria is the production of bacteriocins, small peptides that can inhibit the growth of closely related microorganisms, but that do not affect the producer strain due to immunity mechanisms [35]. The inhibition of closely related microorganisms confers a competitive advantage to eliminate bacteria that may compete for similar nutrients or adhesion sites, for example. Coagulase-negative *Staphylococcus* (CoNS) species, predominant members of the skin microbiome, are known to produce a wide range of bacteriocins. O’Sullivan and colleagues studied bacteriocin producers from different sites of the skin and found 21 bacteriocins with activity against common human skin pathogens, including methicillin-resistant *Staphylococcus aureus* (MRSA) [36]. Bacteriocin producers were predominantly members of the *Staphylococcus* genus, including *S. capitis*, *S. hominis*, *S. epidermidis*, *S. simulans*, and *S. warneri*. Among them, *S. epidermidis*, *S. hominis*, and *S. capitis* are the major staphylococci found colonizing the skin of healthy individuals, considering children, adults, and seniors, and they have been considered ubiquitous commensals of human skin [13]. A wide range of bacteriocins produced by *S. epidermidis* have already been described, such as Epidermin, Pep5, and Epilancin K7, which can affect different Gram-positive bacteria, including MRSA, other coagulase-negative *Staphylococcus* species (CoNS), and even *S. epidermidis* strains that do not produce these bacteriocins [14]. Another CoNS, *S. capitis*, was reported to produce a bacteriocin called capidermicin, which showed antimicrobial activity against all of the tested Gram-positive bacteria (*Lactococcus latis*, *S. aureus*, *Staphylococcus intermedius*, *Staphylococcus pseudintermedius*, and *Micrococcus luteus*) [21]. Recently, Fernandéz-Fernandéz and colleagues (2022) tested the production of bacteriocin-like inhibitory substances (BLIS) from staphylococci of different sources. Both coagulase-positive and -negative *Staphylococcus* isolates were found to produce BLIS. In addition, 13 BLIS-positive isolates showed antimicrobial activity against several bacterial species, including methicillin-resistant *S. pseudintermedius* (MRSP), MRSA, vancomycin-resistant enterococci (VRE), *Listeria monocytogenes*, and *Clostridium perfringens* [37].

Another study where strains isolated from healthy skin of people of different ages confirmed the notion that skin commensals harbor a wide range of antimicrobial molecules [13]. In this study, strains of the most commonly isolated *Staphylococcus* species, *S. epidermidis* and *S. hominis*, showed antimicrobial activity against different skin pathogens. The authors identified a bacteriocin produced by an *S. hominis* strain, named MP1, with broad activity against Gram-positive species, including multidrug-resistant strains, such as community-, hospital-, and livestock-associated MRSA strains, penicillin-resistant streptococci, VRE, and methicillin-resistant CoNS [13]. Furthermore, a variant of Nisin, a bacteriocin already approved and used for food preservation, Nisin J, was shown to be produced by *S. capitis* isolated from skin [22]. The authors reported the antimicrobial activity of Nisin J against human skin pathogens, including staphylococci, streptococci, and *C. acnes*, which can also behave as an opportunistic pathogen.

The ecological role of these bacteriocins in shaping the skin microbiome and impacting colonization by potential pathogens in vivo is not well established. This discussion has been raised in the context of the nasal microbiome. This site is colonized by a wide range of staphylococci, with most strains harboring bacteriocin-production genes [38]. At the same time, it is also a major site for *S. aureus* colonization, suggesting that the sole presence of bacteriocin producers is not a significant obstacle for *S. aureus* establishment [6]. One of the hypotheses to explain this apparent discrepancy is that *S. aureus* colonization would not necessarily be impacted because this species can also harbor genes encoding bacteriocins, some of which are similar to those produced by *S. epidermidis*. Therefore, resistance genes for self-produced bacteriocins would confer cross-resistance to CoNS bacteriocins [6]. On the other hand, in a screening of anti-*S. aureus* activity by bacterial isolates from the skin of atopic dermatitis patients and healthy individuals, it was found that most CoNS isolates from healthy individuals were able to inhibit *S. aureus* growth, whereas isolates from atopic dermatitis patients showed lower activity [39]. The authors of this study also observed a lower frequency of anti-*S. aureus* activity by CoNS isolates from a healthy individual colonized by *S. aureus*, suggesting that the presence of strains that produce anti-*S. aureus* bacteriocins is relevant to colonization resistance against this pathogen. In fact, a specific *S. hominis* strain with anti-*S. aureus* activity, when applied to a pigskin model of *S. aureus* colonization at a similar density of what is found in human skin, resulted in a significant decrease in *S. aureus* colonization [39]. Therefore, these data indicate that, indeed, antimicrobial activity by commensal staphylococci may play an important role in controlling *S. aureus* skin colonization in vivo.

*S. lugdunensis*, an important colonizer of human skin, has also been described as a producer of lugdunin, a thiazolidine-containing cyclic peptide antibiotic that prevents colonization by *S. aureus* [24]. Treatment of mice with lugdunin produced by an *S. lugdunensis* isolated from the nares led to a reduction or eradication of *S. aureus* on the surface and in the deeper layers of the skin [24]. More recently, it was demonstrated that pretreatment of human keratinocytes or mouse skin with lugdunin associated with *S. epidermidis* conditioned medium, which can induce production of antimicrobial peptides, promoting host innate defenses, led to a significant reduction in *S. aureus* colonization [40]. These findings show not only a direct inhibitory effect of molecules produced by the microbiome but also how these molecules can act synergistically with host defenses to protect against skin pathogens.

Molecules with antibacterial activity produced by skin commensals can affect not only exogenous pathogens but also other skin commensals, helping to shape the structure of the skin microbiome community. For example, it has been reported that *C. acnes* has antibacterial activity against *S. epidermidis* [41]. In this work, authors investigated the antagonistic interactions between *C. acnes* and *S. epidermidis*, using strains that were isolated from healthy and acne-affected skin. They found a genomic island in the accessory genome of *C. acnes* that encodes a thiopeptide similar to berninamycin A, a cyclic thiopeptide antibiotic first isolated from *Streptomyces bernensis.* Berninamycin A inhibits protein biosynthesis in Gram-positive bacteria by binding to ribosomal subunits [42]. Claesen and colleagues (2020) named the thiopeptide produced by *C. acnes*, cutimycin, and showed that this thiopeptide had potent in vitro activity against *Staphylococcus*, but not Actinobacteria, from skin. It has also been reported that *C. acnes* can confer host resistance against *S. aureus* in a *C. elegans* model [43]. A different class of molecules with antimicrobial activity known to be produced by skin commensal staphylococci are the phenol soluble modulins (PSMs). PSMs are amphipathic α-helical peptides found in virtually all staphylococci, and many functions have been attributed to them. PSMs have surfactant activity and some of them are known to be able to lysis host cells, such as neutrophils, through pore formation in the membrane [44]. PSM-γ and PSM-δ, produced by *S. epidermidis*, seem to cause not only host damage but also affect bacterial cells, as antimicrobial activity has been demonstrated against *Streptococcus pyogenes*, also known as group A streptococci (GAS), and *S. aureus*, both important human skin pathogens [45]. The presence of *S. epidermidis* PSM-δ on the surface of healthy skin has been demonstrated. Additionally, PSM-δ seems to interact with host immune cells, improving their response against GAS infection [15]. PSMs produced by another skin *Staphylococcus* species were also demonstrated to have antimicrobial activity. O’Neill and colleagues performed a screening looking for antimicrobial activity against *C. acnes*, which, besides being a common skin commensal, can also cause different types of infections. The authors found four PSMs produced by an *S. capitis* strain that act synergistically against *C. acnes* [23].

Another mechanism through which skin commensals can affect the growth of other microorganisms is through fermentation, mainly due to the activity of short-chain fatty acids (SCFAs), major end products of this process [46]. In this regard, *C. acnes* is known for its ability to ferment carbohydrates to propionic acid. Although this SCFA displays antimicrobial activity, *C. acnes* has a high tolerance to it and is able to maintain growth in the presence of this SCFA [47]. Shu et al. (2013) showed for the first time the impact of SCFAs produced by commensal *C. acnes* against *S. aureus* USA300, a community-associated MRSA lineage that is the predominant cause of skin and soft tissue infections in the USA [48]. SCFAs produced through fermentation of glycerol, a carbon source naturally found on the skin [49], decreased USA300 colonization of skin lesions, due to a decrease in intracellular pH, affecting bacterial growth [26]. This seems to be a mechanism used by *C. acnes* to outcompete *S. aureus*, conferring it some advantage, as it is known that both species can ferment the same carbon source, competing when in the same niche [50]. There are also reports in the literature that SCFAs produced by *C. acnes* have antibiofilm activity. SCFAs present in the supernatant of *C. acnes* cultures have been shown to inhibit biofilm formation by *S. epidermidis* [27]. It has been also shown that SCFAs produced by *S. epidermidis* from glycerol fermentation can inhibit *C. acnes* growth [16]. Acne lesions seem to facilitate the overgrowth of *C. acnes* due to the anaerobic microenvironment created, but *S. epidermidis* and other members of the skin microbiota co-exist at these sites [51]. So, Wang and colleagues hypothesized that *S. epidermidis* fermentation would be triggered by anaerobic conditions, and that the ensuing production of SCFAs would affect *C. acnes* growth within acne lesions [16]. Besides the effect of SCFAs produced by *S. epidermidis* during glycerol fermentation against *C. acnes*, *S. aureus* has also been shown to be affected [17,52]. Interestingly, in a study that evaluated the fermentative abilities of skin commensals, all 155 staphylococci strains, belonging to different species, were able to ferment glycerol [53]. Although the production of SCFAs with antimicrobial activity was not evaluated, this observation raises the question of whether glycerol fermentation products produced by all these species could help shape the skin microbiome.

Because Gram-positive species dominate the bacterial community at the skin [4], most studies about microbial interactions in the skin are performed with Gram positives. Myles and colleagues (2016), motivated by a previous study that showed differences in the presence of Gram-negative bacteria in atopic dermatitis patients and healthy individuals [54], investigated the potential of culturable Gram negatives isolated from the skin against *S. aureus*, a major concern in atopic dermatitis patients. They showed that some of the isolates, particularly *Pseudomonas aeruginosa* and *Roseomonas mucosa*, were able to impact *S. aureus* growth in vivo [30,31,32]. These results highlight the relevance of Gram-negative species in shaping the microbiota community in vivo.

*Lactobacillus*, a frequent and important commensal of the vaginal microbiome, is also present in the skin of healthy individuals [34,55,56]. Similarly to what is known about the vaginal microbiome, these bacteria appeared to also have a positive impact on skin health. Lebeer and colleagues (2022) recently evaluated the supernatant of different species of Lactobacillaceae (*Lacticaseibacillus rhamnosus*, *L. plantarum*, and *Lactiplantibacillus pentosus*) and showed that all strains tested were able to inhibit the growth of *C. acnes* and *S. aureus* [29].

Most studies regarding microbial interactions on the skin have focused on the bacterial components of this community, mainly because they are the most abundant group found on the human skin [4]. But recent studies have shown inter-kingdom interactions that may also affect skin microbial communities. *Malassezia globosa* is one of the most common fungal species found in the human skin and it can produce the protease MgSAP1 (*Malassezia globosa* Secreted Aspartyl Protease 1), which was shown to have activity against *S. aureus* virulence factors [34]. This protease cleaves a major virulence factor of *S. aureus*, protein A (Spa), and affects biofilm formation by this pathogen. Analyses of healthy volunteers’ skin samples showed that MgSAP1 was expressed in almost all of them, and the expression levels were similar across different skin sites, suggesting a biological role for this protease on healthy skin. The role of MgSAP1 on *S. aureus* establishment in vivo remains unclear. Nevertheless, it is worth noting that *M. globosa* abundance has been reported to be significantly lower in the skin of atopic dermatitis patients, compared to healthy individuals [57], and the skin of these patients are heavily colonized by *S. aureus* [58]. Protease production is a known mechanism used during microbial antagonistic interactions. Studies to understand competitive interactions at human nares showed that *S. epidermidis* production of Esp, a protease able to disrupt the *S. aureus* biofilm, resulted in the eradication of this pathogen from this site [18,19]. Recently, a study evaluating a *S. pyogenes* strain showed that production of SpeB, a streptococcal protease, can disrupt *S. aureus* USA300 biofilms through the cleavage of the SdrC adhesin [33]. Although *S. pyogenes*, as *S. aureus*, is an opportunistic skin pathogen, it can also be found colonizing healthy skin and occupying similar niches to *S. aureus* [59]. Therefore, production of this protease might serve as a competitive advantage for *S. pyogenes* in the skin.

Studies such as the ones on the Esp protease, which showed that by disrupting *S. aureus* biofilms, the protease prevents the establishment of the pathogen, are examples of how the microbiome can affect pathogens not only by affecting growth but also their virulence attributes. The production of these antibiofilm molecules can be an important pathway for bacterial competition on the skin. In a study carried out by our group, molecules present on culture supernatants of *S. epidermidis* showed antibiofilm activity against MRSA and MSSA, without any impact on growth. Furthermore, these molecules were able to disrupt pre-established *S. aureus* biofilms and reduced the antibiotic concentration required to eliminate them in vitro [20]. Several hundreds of genes were differentially expressed in the presence of these molecules, including not only biofilm-associated genes but other important *S. aureus* virulence determinants. More recently, we have also shown that molecules produced by *C. acnes* inhibit the formation of *S. lugdunensis* and *S. hominis* biofilms. This activity was specific to biofilms, as no impact on growth was observed [28]. Since *C. acnes*, *S. lugdunensis,* and *S. hominis* are part of our skin microbiome, we hypothesized that these antibiofilm molecules may have an impact on colonization of specific niches by these species. More recently, Abbott and colleagues (2022) also described *C. acnes* antibiofilm activity against *S. aureus*. The authors observed that *C. acnes* sterile supernatants reduced the biomass of *S. aureus* biofilms, and these showed greater susceptibility to antibiotics [60].

Most studies about skin microbial interactions have been directed to elucidate competition mechanisms, but interactions found in complex communities are not always of antagonistic nature. Kwaszewska and colleagues (2014) evaluated the enzymatic arsenal of CoNS and *Corynebacterium* species and their ability to survive in response to skin environmental conditions, such as low pH, osmotic stress, and low nutrient availability. The authors found that these bacteria may benefit from each other, supporting colonization levels at a ratio that is important to maintain the microbial balance of the normal skin [53]. They showed that most *Corynebacterium* spp. lack proteinase, phospholipase, and saccharolytic activity, differently from CoNS strains, which can potentially explain the dominance of CoNS colonizing the skin. They suggested that corynebacteria could benefit from the products resulting from enzymatic activity by CoNS to survive in the skin, in a cross-feeding mechanism [53]. Synergistic interactions are also found in skin diseases. Seborrheic dermatitis (SD) lesions have been associated with *Malassezia* proliferation on the skin, due to its increased amount at the lesions, and this is supported by the fact that antifungal treatment improves the clinical condition [61]. However, there is also a high level of *S. epidermidis* colonization of SD lesions, and treatment with topical antibiotics also relieves symptoms. This suggests that there is a role for *S. epidermidis* in the pathogenesis of the disease as well [62]. In a first step to evaluate the interaction between the two species involved in SD pathogenesis, Han and colleagues showed that *Malassezia furfur*, the main fungal species found among SD patients in China, promotes the growth of *S. epidermidis* under conditions of lipid deficiency, commonly found in SD lesions [63]. They attributed the *S. epidermidis* overgrowth to the pH increase caused by *M. furfur* and hypothesized that other bacteria that could benefit from this pH increase would be inhibited by *S. epidermidis* through different competition mechanisms. Together, these data indicate that the skin community is a complex environment, with various types of intercellular interactions that need to be further investigated if we are to truly comprehend the role of skin microbial communities in health and disease (Figure 1).

## 3. Bacterial Interactions in the Skin That Affect the Regulation of Quorum Sensing Systems

Various mechanisms can be used by microorganisms to sense environmental stimuli and shape their behavior accordingly, through the modulation of gene expression. External stimuli sensed by bacteria often include non-specific cues, such as pH, temperature, and osmolarity, but can also include molecules produced by other microorganisms, which can act as chemical cues and signals involved in cell–cell signaling [64]. Chemical communication can occur between members of the same or different species, being classified as intra- and interspecific communication, respectively. Interspecific communication can happen even between different kingdoms, as discussed herein.

One of the major communication mechanisms used by bacteria is called quorum sensing (QS), a mechanism that many bacteria use to shape community behavior in response to cell population density [64]. Generally, during bacterial growth, a molecule termed autoinducer is produced and secreted into the environment. When the autoinducer reaches a critical concentration, due to the high cell density, a receptor senses the molecule and activates the system [65].

QS is used by bacterial cells to synchronize their behavior at a community level and regulate the expression of genes that are important in that particular environment (high vs. low density). In bacterial pathogens, the activation of QS is usually associated with the production of a wide range of virulence factors, such as bacterial toxins [66]. Therefore, the system was originally described as a regulatory mechanism used for communication between cells of the same species. However, since then, many studies have shown that molecules produced by cells from different species (and even kingdoms) can be sensed through QS, affecting these cells in different ways [64] (Table 2).

*Staphylococcus* spp. harbors one of the most well-studied QS systems, the Agr system, and interspecific signaling through Agr seems to be important for shaping the skin community and also have an impact during skin diseases [68,69,70,71,73]. The *agr* locus consists of two operons (RNAII and RNAIII) controlled by two promoters (P2 and P3, respectively). The RNAII transcript unit encodes four genes of the system: *agrD*, which encodes the autoinducing peptide (AIP) precursor; *agrB*, which encodes the transmembrane endopeptidase that processes and exports the AIP precursor; *agrC*, encoding the transmembrane histidine kinase that senses the AIP; and *agrA*, which encodes a response regulator that, when phosphorylated, binds to P2 and P3 to autoregulate RNAII and activate the transcription of RNAIII. RNAIII encodes *hld*, which encodes a delta-hemolysin, as well as a small regulatory RNA that is responsible for the regulation of many virulence factors [74].

The Agr system is present in all staphylococci and differences along the *agrBDCA* hypervariable region, which consists of the 3′end of *agrB*, *agrD,* and the 5′end of *agrC*, resulting in different structures and defines the Agr type (or group) of each strain [74]. The variability of Agr types was initially described in *S. aureus*, which has four Agr types [75]. It has been shown that AIP from each Agr-type needs their specific AgrC sensor to activate the system, and, importantly, a non-cognate AIP can interact with AgrC and block the system [76]. This cross-inhibition, or Agr interference, between members of *Staphylococcus* has been described as an important mechanism that could confer some protection to the skin. The first inter-species cross-inhibition was described by Otto et al., who showed that the AIP from *S. epidermidis*, one of the major commensal skin species, was able to inhibit the Agr of *S. aureus*, repressing virulence factors regulated by QS [73]. Later, the same group showed that AIP produced by one *S. epidermidis* strain was able to inhibit *S. aureus* Agr types I, II, and III but not Agr type IV [67]. It has already been shown that the *S. aureus* Agr type IV has evolved from type I, with only one amino acid modification [77], and interestingly, *S. aureus* Agr type IV is the only one capable of inhibiting the *S. epidermidis* Agr system. Therefore, the authors suggested that this evolutive process might have occurred under the selective pressure of competition between both species in the skin, as *S. aureus* Agr type IV isolates are often involved in skin infections [67,78]. An interesting hypothesis that emerged was that the ability of *S. epidermidis* AIP to cross-inhibit almost all *S. aureus* Agr types, resulting in an advantage for competition within this niche, resulting in the high prevalence of *S. epidermidis* on the skin. The number of studies about a possible antagonistic crosstalk between *S. aureus* and other common staphylococci members of the skin microbiota has increased over the years.

Many studies have shown that an active Agr system is crucial for the development of *S. aureus* skin infections [79,80,81], and factors regulated by this system, such as PSMα, are also involved in the induction of epidermal barrier damage in atopic dermatitis [68], highlighting the importance of understanding the interactions that might occur within skin microbial communities. Recently, Williams and colleagues (2019) confirmed that *S. epidermidis* Agr type I AIP can inhibit *S. aureus* Agr type I, the most frequently observed type in *S. aureus* atopic dermatitis isolates [82,83,84]. *S. epidermidis* AIP from Agr types II and III, however, did not have the same effect. By correlating the abundance of *S. epidermidis* Agr type I with *S. aureus* presence through metagenomic data, authors showed that *S. epidermidis* Agr type I was less abundant in AD subjects with higher disease severity. The same study showed not only that *S. epidermidis* might be involved in antagonizing *S. aureus* through Agr cross-inhibition in AD but also *S. hominis*, *S. warneri*, and *S. capitis*, which displayed potent inhibitory activity against *S. aureus* Agr [68].

As described previously, there is a robust body of evidence in the literature showing that skin commensals produce a wide range of molecules that can directly inhibit the growth of pathogens, such as bacteriocins. More recently, the effect of commensal-secreted molecules on signaling systems, inhibiting pathogens without any effect on growth, has been gaining attention. Competition experiments using a murine model with *Staphylococcus caprae*, another CoNS species than can be found in the skin of healthy individuals, showed that its AIP inhibited the *S. aureus* QS system, preventing not only damage during infection but also colonization by *S. aureus* [69]. Remarkably, *S. caprae* AIP was shown to inhibit all *S. aureus* Agr types [69]. The mechanisms involved in *S. aureus* Agr interference and its impact on skin colonization is not well established. However, some data point to the importance of the Agr system on *Staphylococcus* sp. skin colonization. For example, it has been shown, using a porcine skin model, that a functional Agr is important for successful colonization of the skin by *S. epidermidis* [85]. Then, Pahariki and colleagues (2017) discussed that considering that the Agr system is present in virtually every *Staphylococcus* sp. tested, it may have the same relevance for different species. Interestingly, it was recently shown that AIP produced by *Staphylococcus simulans*, a rare CoNS skin commensal, can inhibit all *S. aureus* Agr types, showing that this phenomenon is not limited to the most abundant species [70]. Use of this molecule on a murine skin infection model showed therapeutic potential, reducing dermonecrosis and skin injury [70]. In a recent paper published by Severn and colleagues (2022), three novel AIP sequences (types IV to VI) identified in *S. hominis* showed the ability to inhibit the *agr* signal in MRSA. Furthermore, they reported that *S. hominis agr*-II was predominant in healthy human skin from study subjects. AIP-II from *S. hominis* was shown to protect against USA300 MRSA in a skin model of dermonecrosis, where mice treated with AIP-II had smaller lesions compared to untreated mice [71].

Agr crosstalk is the best studied example of signaling interference between staphylococci species, but AIP may not be the only molecule that can interact and inhibit the system. In a recent paper from our group, molecules produced by an *S. epidermidis* healthy skin isolate were able to regulate a wide range of *S. aureus* genes, including genes related to the Agr system [20]. Genes that encode the Agr system and genes regulated by this quorum sensing system were downregulated when *S. aureus* was grown in the presence of *S. epidermidis* molecules, suggesting Agr interference. The *S. epidermidis* strain was characterized as Agr type II, and as described above, some studies have shown that only *S. epidermidis* Agr type I is able to inhibit the *S. aureus* Agr system, suggesting that other molecules produced by this *S. epidermidis* strain are responsible for the inhibition of *S. aureus* Agr [20].

Staphylococci species are not the only commensal skin members that have been shown to inhibit *S. aureus* Agr. Experiments of *S. aureus* co-cultures with *Corynebacterium striatum* showed a shift from the expression of virulence factors to the expression of genes associated with commensal behavior in *S. aureus* [72]. Among the virulence genes modulated by co-culturing, there were toxins and other genes regulated by the Agr system, indicating that *C. striatum* molecules might modulate Agr expression. When evaluating the effect of *C. striatum* cell-free conditioned medium on Agr activation, a similar inhibition was seen in *S. aureus* Agr types I, II, and III, but an Agr type IV isolate was not affected. The authors showed that not only *C. striatum* but other *Corynebacterium* spp. were able to inhibit *S. aureus* Agr expression. In addition, results obtained in vivo, using a subcutaneous abscess model, showed that *S. aureus* was less abundant when co-inoculated with *C. striatum*, suggesting that in the presence of this commensal, *S. aureus* seems to be a less successful pathogen [72].

## 4. Biotechnological Applications of Products Obtained from Skin Interactions

Competition between microorganisms in complex microbial communities resulted in the evolution of compounds that are able to inhibit different microorganisms, including some relevant human pathogens. Thus, microbial interactions have been widely exploited as a source of potential therapeutic compounds since the discovery of penicillin, with most of the studies focused on soil community interactions [86]. As human microbiome studies have arisen, microbiome interactions have emerged as a potential source of new compounds [87]. Although most efforts to date have focused on the gut microbiome, there is a wide range of molecules produced by skin commensals with different inhibitory mechanisms and great potential against human pathogens (Table 3). Skin commensals can produce a wide range of bacteriocins, a class of molecules that is already used in different fields for bacterial growth control. For example, Nisin J, recently isolated from an *S. capitis* strain from human skin [22], is a natural variant of Nisin, one of the oldest known bacteriocins, which has been used in food preservation since 1953 [88]. Nisin has also been researched for both human and veterinary clinical purposes, not only due to its antimicrobial activity but also because it can modulate immune responses to pathogens [89,90]. Interestingly, the structure of Nisin J harbors modifications that seem to enhance its bioactivity compared to Nisin, probably due to its constant evolution through interactions with various skin pathogens [22]. Oftentimes, isolation of a compound with great activity in vitro does not translate into a good option for clinical treatments due to the formulation and drug delivery issues. Liu and colleagues (2020) described an *S. hominis* strain able to produce a bacteriocin already known to be produced by other bacterial species (MP1). MP1 has been described as an antibacterial, antiviral, and antitubercular compound, but with poor solubility for further effective formulation. So, the authors developed an MP1 nanoparticle and showed its efficacy in treating *S. aureus* local and systemic infections [13].

Production of antimicrobial molecules by skin commensals highlights not only the possibility of isolating new antimicrobial molecules for drug development but also the potential use of a live producer strain for therapeutic purposes. The use of a live strain may have advantages due to the constant production of the therapeutic compound, maintaining the activity at the desired site of delivery. Some treatments using live bacterial strains obtained from the skin microbiota have been proposed for skin diseases. In one study, authors isolated CoNS strains from the skin of healthy individuals and searched for anti-*S aureus* activity. By doing so, they found a *S. hominis* strain (A9) that produces an antimicrobial peptide, a lantibiotic, against *S. aureus*. This isolate was able to reduce *S. aureus* skin colonization in both a pig and a mouse skin infection model. Furthermore, the authors investigated if the application of a formulated cream using live CoNS strains with antimicrobial activity in the skin of atopic dermatitis patients would affect *S. aureus* colonization in these patients [39]. This formulation was prepared with *S. hominis* and/or *S. epidermidis* isolated from the non-lesional skin of the AD patient, in an autologous microbiome transplant model, in which the strains were confirmed as antimicrobial producers by whole-genome sequencing. They showed that one dose of the CoNS formulation was able to reduce the *S. aureus* burden in lesional skin of the patients [39]. Another aspect of using live cells and not only the antibacterial active compound is that these bacteria may play different roles and act through different mechanisms, improving different aspects of the disease. Myles and colleagues (2016) showed that applying a strain of *R. mucosa* isolated from the skin of a healthy volunteer resulted not only in the control of *S. aureus* burden in the lesions but also enhanced barrier function and innate immunity activation in mouse models. Interestingly, when the supernatant or dead cells of the same strain of *R. mucosa* were tested, they did not provide the same results as the live strain, suggesting the requirement of the live strain for the development of further treatments [30]. Clinical results have already shown efficacy of the treatment with *R. mucosa* in adults and children with AD, showing beneficial results regarding *S. aureus* burden as well as enhanced barrier and immune responses, confirming the previous data [31]. More recently, clinical data obtained from the treatment of children at the most common age group for AD with *R. mucosa* revealed promising results and demonstrated its role in epithelial skin barrier repair [32]. *Lactobacillus* has also recently been tested as a probiotic for acne patients. Lebeer and colleagues (2022) made a topical formulation containing *Lactobacillus* strains and tested it on acne patients [29]. They observed that this formulation was able to reduce the inflammatory lesions and modulate the microbiome. Furthermore, they reported that a reduction in inflammation was observed up to 4 weeks after ending treatment.

As discussed above, metabolites resulting from fermentation by skin commensals may be useful against skin pathogens, and this concept has been raised as a new approach for the treatment of skin diseases. However, there are some issues that make this approach unlikely to be useful for therapeutic purposes. SCFAs usually have a short half-life and are needed in high concentrations to act as antimicrobials. Therefore, high doses would be required for in vivo efficacy, which would result in a higher pH, affecting the host cells. Besides that, SCFAs, such as butyric acid, are characteristically malodorous. Researchers have been trying to develop pro-drugs based on SCFAs to solve these issues. Traisaeng and colleagues (2019) synthesized a derivative molecule from butyric acid, an SCFA produced by *S. epidermidis* through glycerol fermentation, that showed anti-*S. aureus* activity in a lower concentration than the original SCFA. The synthesized molecule, BA-NH-NH-BA, reduced *S. aureus* colonization and improved the production of pro-inflammatory interleukin-6 in a mouse model. An esterified molecule derived from propionic acid, with increased half-life, has also been developed and similar results against *S. aureus* were obtained in vitro [47]. However, studies showing in vivo efficacy, half-life, and dose concentrations remain to be performed.

Another approach to exploit the antimicrobial activity of SCFAs is applying the fermentative source directly to the skin surface. However, different microorganisms can ferment the same source, causing the overgrowth of both beneficial and harmful microbes. A treatment hypothesized and proposed for acne dysbiosis, when there is a disbalance with the overgrowth of *C. acnes* in the microbiome, involves the use of sucrose as a source and not glycerol. SCFAs produced by *S. epidermidis* through glycerol fermentation can suppress *C. acnes* growth, but *C. acnes* can also ferment this source. So, Wang and colleagues (2016) suggested the use of sucrose as a selective fermentation initiator (SFI) to intensify *S. epidermidis* fermentation but not *C. acnes*. The presence of sucrose during co-infection in intradermal mouse models resulted in lower *C. acnes* colonization and inflammation [91]. Studies evaluating the sucrose fermentative potential of other skin commensals would be worthy to understand the impact of this treatment in skin microbiome balance. Findings such as this may be useful for the development of an antibiotic adjuvant, which would potentiate its antibacterial effect. A similar approach of using an SFI was shown to suppress the overgrowth of the fungus *Candida parapsilosis*, usually associated with seborrheic dermatitis, by production of SCFAs through *Staphylococcus lugdunensis* fermentation [52]. The synthesized mPEG-PCL polymer [methoxy poly (ethylene glycol)-b-poly (ε-caprolactone)] worked as a fermentative source for *S. lugdunensis* production of acetic and isovaleric acids. Acetic acid and its prodrug (Ac-DEG-Ac) were able to suppress *C. parapsilosis* growth and when tested in human dandruff suggested a broad spectrum of anti-fungal activity [52].

Interactions between microorganisms in complex communities result not only in molecules that can affect the growth of others but also molecules that affect the signaling systems and virulence factors, as discussed herein. Over the last years, the emergence of multidrug-resistant pathogens has been a worldwide public health issue, resulting in urgency for the development of new therapeutic approaches. Antivirulence therapies emerged as an alternative, since these compounds only affect virulence but do not cause growth inhibition. Therefore, these compounds would pose a reduced selective pressure for the emergence of resistant strains [92]. Moreover, antivirulence compounds act against specific virulence factors, and therefore, at least in theory, would have a lesser impact on the host microbiota compared to antibiotic treatments [92]. It has also been suggested that antivirulence compounds could be useful to recover antibiotics that otherwise would no longer be effective against resistant strains, potentiating their activity when used in combination. Significant research effort has been applied on molecules that target quorum sensing and other signaling systems in order to target master virulence regulators, affecting a wide range of pathogens’ virulence factors. Within the skin environment, most studies focused on the inhibition of the *S. aureus* quorum sensing system, since *S. aureus* is one of the major skin pathogens, and the Agr system seems to be crucial for its virulence. In vivo assays in murine models of dermonecrotic and cutaneous injury have shown Agr interference as a very promising therapeutic approach against *S. aureus* infections. Both topic application of an AIP that can inhibit *S. aureus* Agr or the application of its producer strain seem to work for reducing the damage caused by *S. aureus* during skin infection [69,70] and improve skin parameters related to atopic dermatitis symptoms induced by *S. aureus* [68]. Agr interference could also be considered for the development of skin probiotic formulations, as data suggest that the presence of certain strains able to inhibit *S. aureus* Agr might affect its establishment [68]. Since *S. aureus* is among the main pathogens with emerging resistance to multiple antibiotics leading to limited treatment options, Agr interference has been raised as a potential alternative for *S. aureus* therapy, but more studies are needed to investigate its clinical implications.

## 5. Future Directions for the Field

Studying the microbiome of the skin presents a specific set of challenges that are found regarding skin microbiome studies. For example, the skin has an uneven microbial distribution, with some areas showing very low levels of colonization. Also, microbes that inhabit deep layers of the skin are harder to sample. Although the use of more invasive methods can allow sampling, this has obvious disadvantages, such as the risk of infection [93]. Furthermore, because it is constantly exposed to the external environment, determining whether a member of the microbiome is a true resident or a transient colonizer remains a challenge in the field. Due to these and other challenges, the skin microbiome is an understudied field when compared to other sites, such as the gut. The combination of constant environmental exposure, resulting in both adverse conditions and constant environmental microbial exposure, as well as poor nutrient availability, makes the skin a highly competitive niche. Therefore, highly efficient microbial strategies for successful colonization of this body site despite the competition must have evolved. As discussed throughout this review, skin microbiome interactions that rely on such strategies can be harnessed as a source of new molecules with therapeutical potential. The fast and steady increase in antibiotic resistance worldwide, combined with the low rate of development of new antibiotics, makes it imperative that new treatment strategies are developed. In this scenario, studies of microbial interactions within microbiomes might shed light on effective mechanisms of microbial competition that could prove useful for the development of new therapeutic approaches. Although many strategies can be envisioned, bioactive compounds with antibacterial or antivirulence activity may be particularly useful and amenable to biotechnological manipulation.

Besides potentially generating compounds with general antibiotic or antivirulence activity that could be used to counter the current issue of antimicrobial resistance, the study of microbial interactions within the skin microbiome may present other translational opportunities. Although the mechanisms involved are still largely unknown, the microbiome has been associated with several skin diseases, such as atopic dermatitis, acne, and psoriasis. These diseases are considered multifactorial and therefore the study of microbial interactions between multiple species, as opposed to the study of single pathogens, as well as of interactions between microbes and host cells, will be key to understanding the driving mechanisms in these skin diseases. Interspecies interactions that prove crucial to the development of disease may become therapeutic targets for inhibition. Conversely, interactions involved in the establishment and maintenance of healthy microbiome communities may be promoted through the use of prebiotics, probiotics, or postbiotics. In the case of probiotics, strategies involving multiple strains or species to reestablish healthy communities will likely be more effective than the use of single-strain alternatives.

Although in this review we have focused on the study of bacterial members of the skin microbiome, there is a growing body of literature on the role of other microbes, such as fungi and protists, as well as viruses on the human microbiome. Here, we briefly discussed a few studies on the mycobiome, which show the role of fungi in skin dynamics and may also provide knowledge for development of new drugs. The skin virome is also a promising field of research that remains vastly underexplored. The majority of viruses that compose the skin microbiome are bacteriophages, which are known to influence and modulate the microbial community [94]. More studies are necessary to better understand the role of these understudied microbiome members in skin health and disease.

## 6. Conclusions

Despite the high number of studies on the skin microbiota published in recent years, this field is still poorly exploited. Because it is constantly exposed to the external environment, several interactions between microorganisms can occur on the skin, and these interactions play an extremely important role in protecting the host against pathogens.

Through the secretion of molecules such as proteases, bacteriocins, and phenol-soluble modulins, the microbiota can not only affect the growth but also attenuate the virulence of other microorganisms. Several skin diseases have already been associated with an imbalance in the skin microbiota, with an exaggerated presence of some microorganisms to the detriment of others, showing the importance of microbial interactions that occur in this site. In addition, microorganisms can sense molecules present in the environment and modify their behavior according to these cues and signals. Chemical communication between microorganisms through quorum sensing has a critical role in interspecies interactions. The Agr system, well studied in *Staphylococcus*, is an example of a quorum sensing system, through which microorganisms can regulate the expression of several genes related to virulence, being important during skin infections. Several molecules produced by the skin microbiota can inhibit signaling systems of pathogens, preventing colonization by the pathogen and damage to the host tissue. Molecules produced by the skin microbiota as well as Agr interference have been raised as a potential alternative treatment for skin infections caused by pathogens such as *S. aureus*. In view of what was presented, the importance of elucidating the interactions that occur in the skin microbiota, not only with the host but also other microbes, is evident, and more in vivo studies are necessary. This can provide new possibilities for therapeutic discovery, which may be important to improve the quality of life of people living with skin diseases.

## Figures and Tables

**Figure 1 antibiotics-13-00049-f001:**
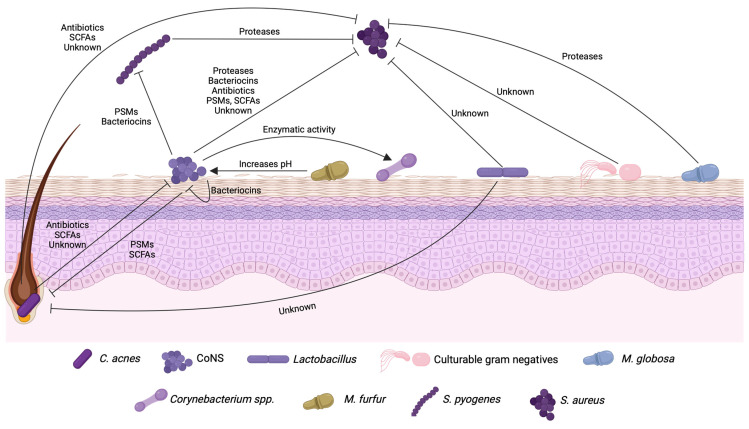
Major players of bacterial interactions in the skin, their targets, and products. “Unknown” refers to molecules that exhibit an inhibitory activity but have not been identified yet.

**Table 1 antibiotics-13-00049-t001:** Bioactive molecules produced by skin commensals that affect the establishment of other microorganisms.

Microorganism	Type of Molecule	Name of Molecule	Effect	Bacteria Targeted	References
*S. epidermidis*	Bacteriocin	Epidermin, Pep5, Epilancin K7	Growth	MRSA, CoNS	[14]
	PSM	PSM-γ, PSM-δ	Growth	*S. pyogenes*, *S. aureus*	[15]
	SCFAs	Acetic acid, butyric acid, lactic acid, succinic acid	Growth	*C. acnes*, *S. aureus*	[16,17]
	Protease	Esp	Biofilm	*S. aureus*	[18,19]
	Unknown	Unknown	Biofilm	*S. aureus*	[20]
*S. capitis*	Bacteriocin	Capidermicin	Growth	*L. latis*, *S. aureus*, *S. intermedius*, *S. pseudintermedius*, *M. luteus*	[21]
	Bacteriocin	Nisin J	Growth	*Staphylococcus* spp., *Streptococcus* spp., *C. acnes*	[22]
	PSM	PSMβ	Growth	* C. acnes *	[23]
*S. hominis*	Bacteriocin	MP1	Growth	MRSA strains, penicillin-resistant streptococci, VRE, methicillin-resistant CoNS	[13]
*S. lugdunensis*	Antibiotic	Lugdunin	Growth	*S. aureus*	[24]
*C. acnes*	Antibiotic	Cutimycin	Growth	MRSA	[25]
	SCFAs	Acetic acid, lactic acid, propionic acid	Growth	*S. aureus*	[26]
		Acetic acid, propionic acid, isobutyric acid, isovaleric acid	Biofilm	*S. epidermidis*	[27]
	Unknown	Unknown	Biofilm	*S. lugdunensis*, *S. hominis*, *S. aureus*	[28]
*Lactobacillus* spp.	Unknown	Unknown	Growth	*S. aureus*, *C. acnes*	[29]
CGN	Unknown	Unknown	Growth	*S. aureus*	[30,31,32]
*S. pyogenes*	Protease	SpeB	Biofilm	*S. aureus*	[33]
*Malassezia globosa*	Protease	MgSAP1	Biofilm and other virulence factors	*S. aureus*	[34]

PSM: phenol soluble modulins; SCFAs: short-chain fatty acids; CGN: culturable Gram negatives; MRSA: methicillin-resistant *S. aureus*; VRE: vancomycin-resistant enterococci; CoNS: coagulase-negative *Staphylococcus*.

**Table 2 antibiotics-13-00049-t002:** Bioactive molecules produced by skin commensals that interfere with signaling between skin microorganisms.

Bacteria	Molecule	Effect	References
*S. epidermidis*	AIP (unknown type)	Inhibition of *S. aureus* Agr types I, II, and III	[67]
	Agr type I AIP	Inhibition of *S. aureus* Agr type I	[68]
	Unknown	Downregulation of *S. aureus* Agr	[20]
*S. caprae*	AIP	Inhibition of all *S. aureus* Agr types	[69]
*S. simulans*	AIP	Inhibition of all *S. aureus* Agr types	[70]
*S. hominis*	AIP-II	Inhibition of *S. aureus* growth	[71]
*C. striatum*	Unknown	Inhibition of *S. aureus* Agr types I, II, and III	[72]

AIP: autoinducer peptide.

**Table 3 antibiotics-13-00049-t003:** Biotechnological applications of products obtained from the skin microbiome.

Products	Application	Benefits	Limitations	References
Nisin J from *S. capitis*	Antimicrobial	Inhibitory activity against a wide range of bacterial targets	Not tested in vivo	[36]
MP1 from *S. hominis*	Antimicrobial	Treatment of *S. aureus* local and systemic infections	No safety tests using probiotic strain	[13]
Topical formulation with live *S. epidermidis* and *S. hominis*	Atopic dermatitis treatment	Highly potent, selectively killed *S. aureus*, and synergized with the human AMP LL-37	A complete catalog of protective bacteria from skin could not be identified	[39]
Topical formulation with live *R. mucosa*	Atopic dermatitis treatment	Enhancement of skin barrier function, innate immune activation, and a reduction in topical steroid requirements without severe adverse events	Small study with children based on historical placebo control data; no data on skin biopsies	[31,32]
Topical formulation with live *Lactobacillus*	Acne treatment	Reduction in inflammatory lesions and microbiome modulation	No information on immunomodulatory mechanisms; viability and activity of lactobacilli	[29]
Derivate molecule from butyric acid	Antimicrobial/ Atopic dermatitis treatment	Ameliorate the production of pro-inflammatory interleukin (IL-6) induced by *S. aureus*, and reduced the colonization of *S. aureus* in mouse skin	Lack of information about mechanisms of action and possible impacts on microbiome	[17]
Derivate molecule from propionic acid	Antimicrobial	Methicillin-resistant *S. aureus* growth inhibition	Not tested in vivo	[47]
mPEG-PCL polymer	Microbiome modulator	Suppression of *C. parapsilosis* growth and prevention of fungal expansion in human dandruff	Not tested in vivo	[52]
Agr interference	Atopic dermatitis treatment	Prevention of skin barrier damage and inflammation	Treatment may promote persistent colonization of *S. aureus* in the skin; no data on stability of the synthetic autoinducer peptide in the skin	[68]

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
