# Peer review of "Microbe Interactions within the Skin Microbiome"

_antibiotics, 2024, doi:10.3390/antibiotics13010049_

Round 1

Reviewer 1 Report

Comments and Suggestions for Authors

 The review article is well organized with up-to-date information. This can be accepted as such.

Author Response

We thank reviewer 1 for his kind review.

Reviewer 2 Report

Comments and Suggestions for Authors

Thank you for requesting my review of the article titled "Microbe interactions within the skin microbiome" authored by Glatthardt et al. The article offers a comprehensive overview of microbe-microbe interactions in the skin microbiome, backed by references and information on biotechnological advancements pertaining to our knowledge of these skin microbes. Overall, I find this to be a well-written article and recommend its acceptance in its current form.

My only suggestion is maybe considering the inclusion of a discussion on bacteriophages, which as we known by now play a significant role in microbe-microbe interactions on the skin. Despite not being classified as microbes, recent research highlights their unique ability to combat microbes like C. acnes. Bacteriophage treatment offers a potential alternative to conventional antibiotic treatments for bacterial infections, and discussing this would be in line with the scope of this journal.

Author Response

Reviewer #2

Thank you for requesting my review of the article titled "Microbe interactions within the skin microbiome" authored by Glatthardt et al. The article offers a comprehensive overview of microbe-microbe interactions in the skin microbiome, backed by references and information on biotechnological advancements pertaining to our knowledge of these skin microbes. Overall, I find this to be a well-written article and recommend its acceptance in its current form.

My only suggestion is maybe considering the inclusion of a discussion on bacteriophages, which as we known by now play a significant role in microbe-microbe interactions on the skin. Despite not being classified as microbes, recent research highlights their unique ability to combat microbes like C. acnes. Bacteriophage treatment offers a potential alternative to conventional antibiotic treatments for bacterial infections, and discussing this would be in line with the scope of this journal.

We thank the reviewer for the suggestion. Although we agree that this is a very interesting subject, we believe that discussing bacteriophage treatment in depth would further lengthen this review. Nevertheless, as suggested by the reviewer, we now have mentioned the role of bacteriophages on microbiome modulation in the “future directions for the field” section.

Reviewer 3 Report

Comments and Suggestions for Authors

In the review paper “Microbe interactions within the skin microbiome”, the authors gave a thorough review of the microbial interactions in skin microbiome including microbe-derived molecules and the quorum sensing system that can affect pathogen infections. The advantage of the paper is that it reviewed the current biotechnological applications of the mechanisms learned from research studies and discussed both the advantages and the limitations of these applications. This reviewer only has a few minor comments.

1.     The paper could benefit from adding more motivations for why reviewing for skin microbiome. Firstly, why skin microbiome should get attention? Secondly, why microbial interactions learned from skin microbiome are important? Are these mechanisms important for breakthroughs in skin disease treatment or these mechanisms can be applied to solve the problem of antimicrobial resistance in general? If it is the latter, then the paper can emphasize more on the urgent needs for antibiotics alternatives due to antimicrobial resistance.

2.     The paper could elaborate on what the “others” mean in Figure 1 as it appeared too many times in this figure. Also, what additional information is provided is Figure 1 than that’s in Table 1? Table 1 seems already covered the information in Figure 1.

3.     The paper should discuss in more details for the future directions of skin microbiome research, including the challenges and the potential use of the knowledge.

4.     The section title “Interactions between microorganisms in the human skin” is broad, and it can even cover the quorum sensing system discussed in the next section. This title can be changed to reflect molecules impact microbial interactions more specially.

5.     The paper can provide a table that summarizes the biotechnological applications and their pros and cons, making it easier for the audience to follow.

Author Response

Reviewer #3

In the review paper “Microbe interactions within the skin microbiome”, the authors gave a thorough review of the microbial interactions in skin microbiome including microbe-derived molecules and the quorum sensing system that can affect pathogen infections. The advantage of the paper is that it reviewed the current biotechnological applications of the mechanisms learned from research studies and discussed both the advantages and the limitations of these applications. This reviewer only has a few minor comments.

We would like to thank reviewer 3 for their work and great suggestions to improve our review.

  1. The paper could benefit from adding more motivations for why reviewing for skin microbiome. Firstly, why skin microbiome should get attention? Secondly, why microbial interactions learned from skin microbiome are important? Are these mechanisms important for breakthroughs in skin disease treatment or these mechanisms can be applied to solve the problem of antimicrobial resistance in general? If it is the latter, then the paper can emphasize more on the urgent needs for antibiotics alternatives due to antimicrobial resistance.

We have now added a section to the manuscript on “future directions for the field” that includes the relevance in understanding microbial interactions in the context of skin diseases. Furthermore, the need for the development of new antibiotics was highlighted, as well as the potential of the skin microbiome as a source of these bioactive molecules.

  1. The paper could elaborate on what the “others” mean in Figure 1 as it appeared too many times in this figure. Also, what additional information is provided is Figure 1 than that’s in Table 1? Table 1 seems already covered the information in Figure 1.

We changed the word “others” in Figure 1 to “unknown”, since these molecules have not been completely characterized and added an explanation in the Figure legend.

Table 1 covers the information illustrated in Figure 1 in greater depth, and it includes the names of the bioactive molecules and their activity, as well as references for each molecule. Therefore, we believe that Table 1 brings extra information and it is necessary for better understanding of the topic.

  1. The paper should discuss in more details for the future directions of skin microbiome research, including the challenges and the potential use of the knowledge.

Challenges related to microbiome studies were added to the new “future directions for the field” section, as well as potential use of knowledge (development of new therapeutical strategies) (line 575).

  1. The section title “Interactions between microorganisms in the human skin” is broad, and it can even cover the quorum sensing system discussed in the next section. This title can be changed to reflect molecules impact microbial interactions more specially.

We changed the titled of the section to “Bioactive molecules produced during microbial interactions in the human skinto better reflect the topic covered.